# Size-dependent deformation behavior in nanosized amorphous metals suggesting transition from collective to individual atomic transport

Naijia Liu [1,2], Sungwoo Sohn[1,3] ✉, Min Young Na[4], Gi Hoon Park[4], Arindam Raj[1], Guannan Liu[1], Sebastian A. Kube [1], Fusen Yuan[5], Yanhui Liu [5], Hye Jung Chang [4,6] & Jan Schroers [1] ✉

The underlying atomistic mechanism of deformation is a central problem in mechanics and materials science. Whereas deformation of crystalline metals is fundamentally understood, the understanding of deformation of amorphous metals lacks behind, particularly identifying the involved temporal and spatial scales. Here, we reveal that at small scales the size-dependent deformation behavior of amorphous metals significantly deviates from homogeneous flow, exhibiting increasing deformation rate with reducing size and gradually shifted composition. This transition suggests the deformation mechanism changes from collective atomic transport by viscous flow to individual atomic transport through interface diffusion. The critical length scale of the transition is temperature dependent, exhibiting a maximum at the glass transition. While viscous flow does not discriminate among alloy constituents, diffusion does and the constituent element with higher diffusivity deforms faster. Our findings yield insights into nano-mechanics and glass physics and may suggest alternative processing methods to epitaxially grow metallic glasses.

Deformation, the response to external stress through a change in configuration, has been a focus of metallurgy for almost 100 years. Plastic deformation in crystalline materials typically utilizes dislocations which slide in slip planes. For amorphous metals, the plastic deformation changes dramatically as a function of temperature and strain rate[1]. At low temperature and high strain rates, deformation is highly localized in narrow shear bands. During such highly heterogenous shear banding controlled deformation, the vast majority of the sample's atoms do not undergo any plastic strain and essentially all plastic strain is carried by the minute fraction of atoms[1-4]. At high temperature and low strain rates, deformation is essentially homogenous. Such dramatic transition in the deformation mechanism within amorphous metals has been explained by a crossover of the relaxation time, the internal time scale enabling structural rearrangements, and the external time scale, set by the applied deformation rate. The relaxation time exhibits a strong temperature dependence and is commonly described by the Vogel-Fulcher-Tamman (VFT) equation[5]. Microscopically, such homogenous viscous flow has been envisioned through a collective movement of atoms, which has been suggested to occur within flow units, also called shear transformation zones

[1]Department of Mechanical Engineering and Materials Science, Yale University, New Haven, CT 06511, USA. [2]Querrey Simpson Institute for Bioelectronics, Northwestern University, Evanston, IL 60208, USA. [3]Yale Institute for Nanoscience and Quantum Engineering, Yale University, New Haven, CT 06511, USA. [4]Advanced Analysis and Data Center, Korea Institute of Science and Technology, Seoul 02792, Republic of Korea. [5]Institute of Physics, Chinese Academy of Sciences, Beijing 100190, China. [6]Division of Nano Convergence, KIST School, University of Science and Technology, Seoul 02792, Republic of Korea. ✉ e-mail: sungwoo.sohn@yale.edu; jan.schroers@yale.edu

(STZs)[6–9], although recent direct experimental evidence suggests atomic homogenous flow[10].

Homogeneous flow enables thermoplastic forming, a unique method in metal processing[11]. A correlation between viscosity ($\eta$), length scale ($d$), strain rate ($v/d$), and applied shear stress ($\tau_s$), $\tau_s = \eta \frac{v}{d}$, is present under low Reynolds numbers and stick conditions of the amorphous metal to the container interface[12]. This relation reveals that $v$ decreases with decreasing $d$ for constant applied stress. Even though molding of amorphous metals has been successfully realized down to small sizes ~10 nm[12–14], this fundamental relation suggests the viscous flow mechanism to become less effective with decreasing length scales. Such predicted behavior points potentially to a change in deformation mechanism for amorphous metals on a small scale.

To advance fabrication and use of amorphous metals, a fundamental and complete understanding of their size- and temperature-dependent deformation is required[15–17]. For example, the size dependence of the deformation of amorphous metals at low temperatures and/or high strain rates has revealed a transition from highly localized and heterogenous deformation to a homogenous mechanism[15,18–20]. This transition is explained by a Griffith-like criterium[18] and has provided valuable insights into energetics of shear band formations[18,21]. However, the study of the effect of size confinement on the deformation under processing conditions of homogenous deformation is still elusive and is the focus of the here presented research.

In this study, we employ nanomolding techniques at elevated temperature region (Fig. 1a) to investigate the size- and temperature-dependency of the deformation mechanism and atomic transport in amorphous metals. Our research reveals a notable size-dependent transition in the deformation behavior of nanosized amorphous metals under compression, occurring at a length scale typically around or below 100 nm. This critical length scale is dependent on both the temperature of deformation and the specific material under study. For sizes larger than the critical length scale, deformation follows the expected description of viscous flow and increasing size resulting in faster deformation rate. However, for sizes below the critical length scale, a substantial deviation from homogeneous viscous flow is observed, exhibiting an increasing deformation rate with reducing size, along with a gradual shifting in the composition of the deformed material. This different scaling behavior of deformation with respect to size indicates a transition in the underlying deformation mechanism, which may be quantitatively explained by the competition and change in dominance between collective atomic transport through homogeneous viscous flow and individual atomic transport via interface atomic diffusion. The experimental findings presented in this work, along with the theoretical framework suggested, provide insights into nano-mechanics and physics of amorphous materials, including but not limited to the deformation mechanisms, the bulk and surface glass transition, breakdown of the Stokes-Einstein relation, and glass structures discussed in this paper.

## Results
### Deformation behavior deviating from homogeneous flow
During nanomolding (Fig. 1a), feedstock materials of amorphous metals are forced into a rigid mold under certain molding pressure ($p$) and temperature ($T$)[12,22,23]. The deformation in nanomolding is confined in a well-controlled nano-geometry defined by the mold, offering a powerful tool to investigate the deformation on the nanoscale as a function of size and temperature. Specifically, as practical examples for amorphous metal molding, we use three distinct glass forming systems: $Ni_{45}Pd_{35}P_{16}B_4$ bulk metallic glass (Ni-BMG), $Zr_{46}Cu_{46}Al_8$ (Zr-BMG), and $Pd_{43}Ni_{10}Cu_{27}P_{20}$ (Pd-BMG), which we mold into cylindrical nanopore arrays with diameters ($d$) ranging from 10 nm to 250 nm. For the fabrication of nanowires, we consider a range of molding temperatures above and below the calorimetric

glass transition temperature ($T_g$ = 317 °C for Ni-BMG, 440 °C for Zr-BMG, and 310 °C for Pd-BMG, Supplementary Fig. 1).

The mechanism of nanomolding has been previously described by the viscous flow of the homogenously deforming amorphous metal. The length of nanowires ($L$) can be determined by a modified Hagen−Poiseuille equation[12]:

$$p = \frac{32\eta}{t}\left(\frac{L}{d}\right)^2 - \frac{4\gamma\cos\theta}{d}, \qquad (1)$$

where $t$ is the molding time, $\gamma$ is the surface tension of the BMG, and $\theta$ is the contact angle of the nanowire-mold interface. At high pressure, the effect of capillary force can be neglected (Supplementary Note 1), and Eq. (1) reveals a relationship, $L/\sqrt{pt} \propto d/\sqrt{\eta}$ (straight line in Fig. 1b and Supplementary Note 2). This relation agrees well with our experimental results for BMGs when the molding temperature $T > T_g$, however, for $T < T_g$ a significant discrepancy between the experimental result and the prediction of the homogeneous flow model (Eq. (1)) is observed. For example, when Ni-BMGs are deformed at $T_g$ and $T_g + 10$ °C (red dots in Fig. 1b), the deformation behavior mostly follows the description of the homogeneous flow model. As a sharp contrast, when molded at $T_g - 10$ °C, $T_g - 20$ °C, and $T_g - 30$ °C (blue dots in Fig. 1b), the deformation behavior of Ni-BMG deviates from the line representing the Hagen-Poiseuille equation, with most of the results exhibit a notable higher deformation rate than predicted by homogeneous flow (Fig. 1b). This observed discrepancy in the deformation behavior of amorphous metals at nanoscales suggests the presence of a mechanism that is distinct from the homogeneous flow currently used to understand deformation in amorphous materials.

### Transiting deformation behavior in scaling experiments
To further investigate this observed discrepancy, we conducted scaling experiments involving the molding length ($L$) and the molding diameter ($d$) of various BMG systems at different temperatures (Fig. 1c–f, data also presented in Fig. 5 and Supplementary Fig. 2). Specifically, in one set of scaling experiments, BMGs are molded at different $d$ under constant temperature, pressure, and molding time. We thus reveal the relation between the deformation rate, as characterized by the molding length ($L$) within the fixed molding time ($t$), with the length scale ($d$). Within the description of homogeneous flow (Eq. (1) and Supplementary Note 1), $L$ is expected to be proportional to $d$ when other molding conditions are fixed.

Figure 1c shows the length characterization of $L$ from SEM investigations for Ni-BMG. The obtained $L$ values are plotted vs. $d$ at a processing temperature of $T_g - 20$ °C (297 °C) (Fig. 1d). A clear transition in the scaling behavior is present at $d$ ~ 60 nm. For larger molding diameters, $d > 60$ nm, the molding length increases linearly with $d$, suggesting a homogeneous flow mechanism. In qualitative contrast, for $d < 60$ nm, $L$ deviates from $L \propto d$ and exhibits an increasing trend as $d$ deceases, resulting in a faster deformation rate than homogeneous flow (Fig. 1c, d and Supplementary Fig. 2). Similar transition phenomena are also observed in nanomolding experiments with Zr-BMG and Pd-BMG (Fig. 1e, f and Supplementary Fig. 2 for details). For example, at a molding temperature of $T_g - 20$ °C, Zr-BMG exhibits a transition at $d$ ~ 120 nm (Fig. 1e) and Pd-BMG shows a transition at $d$ ~ 50 nm (Fig. 1f).

The existence of this transition in the here considered different BMG systems highlights a size-dependent mechanism in the deformation of amorphous metals, which challenges the previously accepted deformation categorization into the homogeneous deformation region[1,12,18]. At large scales, it agrees with the widely accepted homogeneous flow-dominated deformation. In contrast, at small scales, we show here that deformation is no longer solely governed by homogeneous viscous flow.

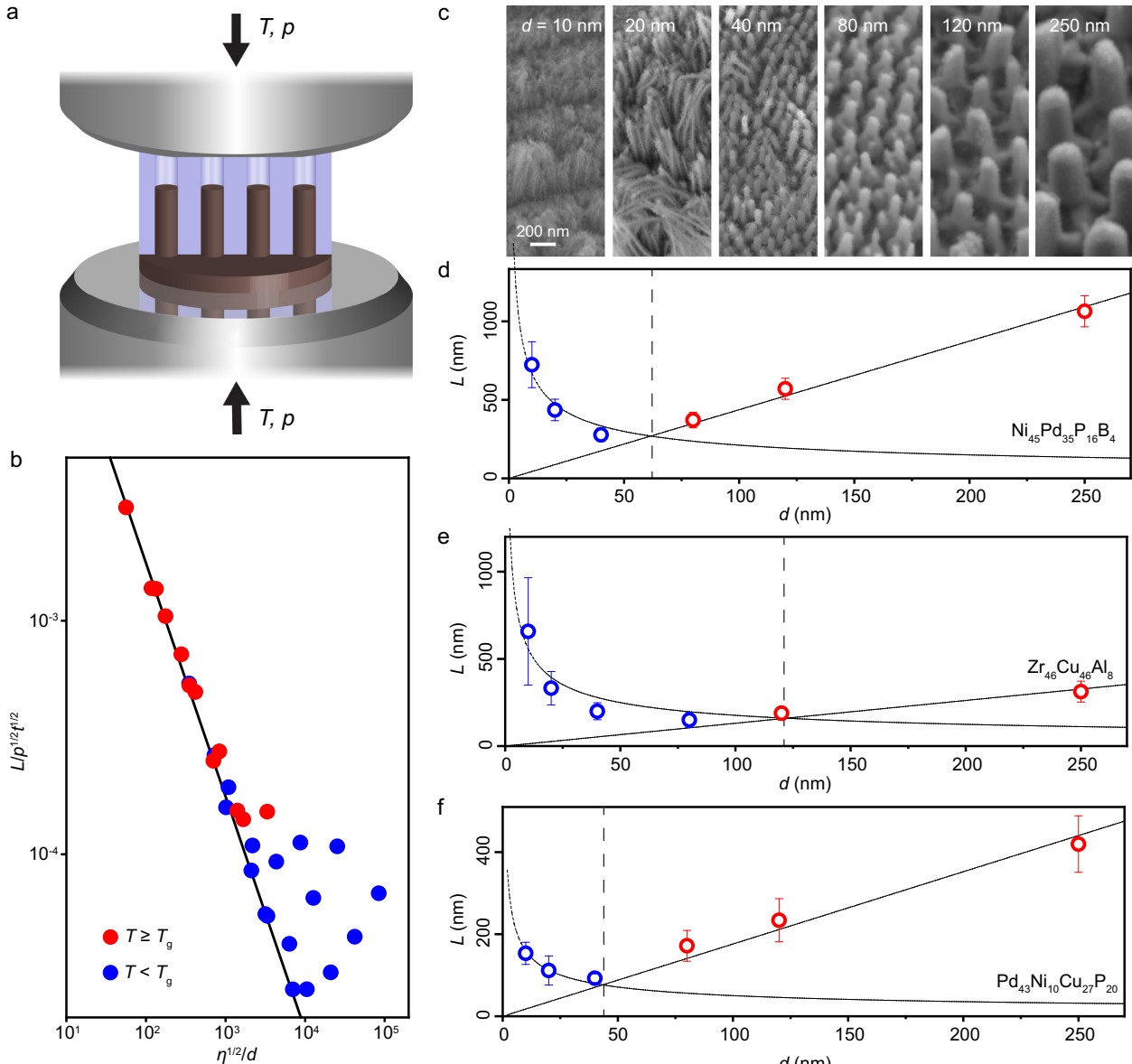

**Fig. 1 | Size-dependent deformation behavior of amorphous metal nanowires. a** A bulk metallic glass (BMG) feedstock is molded into nanocavity arrays under an applied pressure ($p$) and temperature ($T$). **b** The deformation behavior of molded BMGs deviates from the viscous flow model (Eq. (1)). At temperatures $T \geq T_g$ (the glass transition temperature), the normalized length ($L/\sqrt{pt}$) exhibits a linear correlation with $\sqrt{\eta}/d$, where $\eta$ is the viscosity and $d$ is the molding diameter, indicating a mechanism dominated by viscous flow. However, when $T < T_g$, the linear correlation is no longer observed, suggesting a different mechanism than

viscous flow (Supplementary Note 2, data is from Ni-BMG). **c** SEM images of molded nanowires used to determine the molding length $L$ in Ni-BMG ($Ni_{45}Pd_{35}P_{16}B_4$). **d** $L(d)$ scaling results at a processing temperature $T = T_g–20$ °C reveal a transition in size-dependent deformation behavior in Ni-BMG. For $d > 60$ nm, BMG deforms faster with increasing size, whereas the deformation rate decreases with $d$ at $d < 60$ nm. **e, f** Similar transition in deformation behavior in Zr-BMG ($Zr_{46}Cu_{46}Al_8$) and Pd-BMG ($Pd_{43}Ni_{10}Cu_{27}P_{20}$) at $T = T_g–20$ °C. Error bars are defined by the standard deviation of more than 10 measurements. Source data are provided as a Source Data file.

## Effects of deformation mechanism on composition and structure

The observed mechanism transition revealed in the scaling experiments also affects the composition distribution within the formed nanowires. To investigate this, we employed focused-ion beam (FIB) and transmission electron microscopy (TEM) equipped with energy dispersive X-ray spectroscopy (EDS) to measure the composition profile of Ni-BMG nanowires. We examined $Ni_{45}Pd_{35}P_{16}B_4$ samples formed through homogeneous flow region ($d = 120$ nm, $T_g – 20$ °C) and outside the homogeneous flow region ($d = 20$ nm, $T_g – 20$ °C). The EDS data of the nanowire formed through homogeneous flow exhibits a uniform element distribution from the feedstock and throughout the nanowire (Fig. 2a, and additional details in Supplementary Figs. 3 and 4). For the

nanowire formed outside of the homogeneous flow region (Fig. 2b), EDS mapping reveals a different chemical composition between the feedstock and the nanowire. Specifically, the composition of Ni and Pd, which is constant within the feedstock, increases along the growth direction of the nanowire for Ni to ~80% while Pd decreases gradually to ~20% (more details in Supplementary Figs. 5 and 6).

Along with the contrast in the composition profile between nanowires formed in and outside of the homogeneous flow region is the contrast in their atomic structures (Fig. 3). We observed different atomic structures for the Ni-BMG samples formed in and outside of the homogeneous flow region which are shown in Fig. 2a, b. Specifically, the nanowire formed through homogeneous flow exhibits an amorphous structure, which is concluded from the halo ring pattern in the

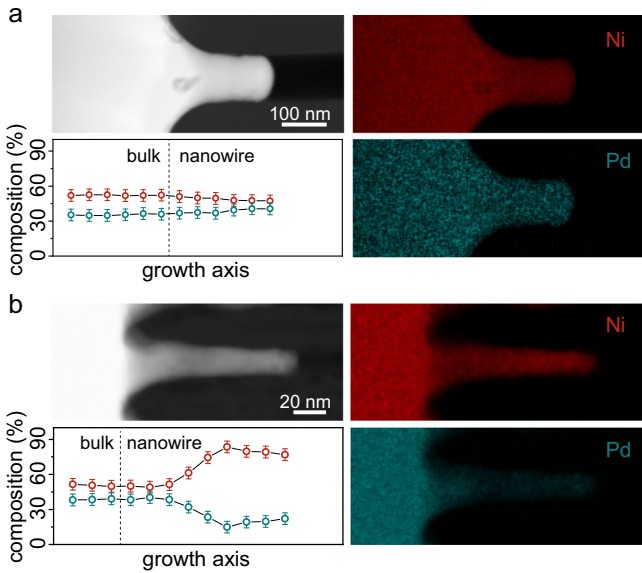

**Fig. 2 | Effect of the suggested mechanism transition on the chemical composition. a** TEM-EDS on a Ni-BMG nanowire deformed through viscous flow, above the transition (molding diameter $d = 120$ nm as shown in Fig. 1d) reveals a uniform composition from the feedstock to the nanowire. **b** TEM-EDS on a Ni-BMG nanowire with molding diameter $d = 20$ nm, deformed below the transition (as shown in Fig. 1d), reveals the composition of Ni gradually increasing along the nanowire. Error bars represent the measurement error of EDS. Source data are provided as a Source Data file.

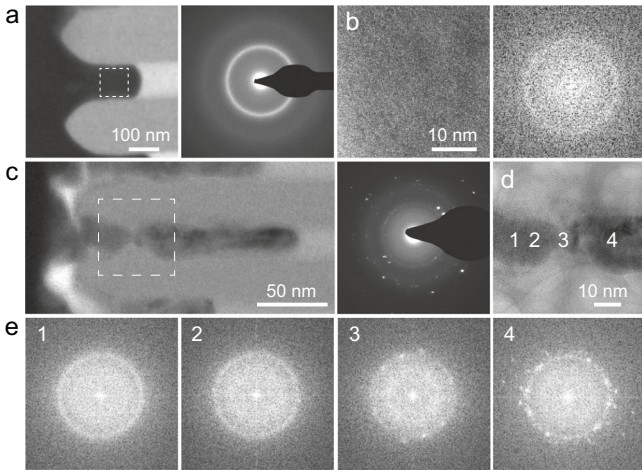

**Fig. 3 | Effect of the suggested mechanism transition on the atomic structure of the nanowire. a**, **b** TEM investigation reveals the amorphous structure of 120 nm Ni-BMG nanowires, formed through homogeneous flow. **a** Overview of a nanowire formed through homogeneous flow where the selected area electron diffraction (SAED) pattern suggests an amorphous structure. **b** HRTEM image and FFT pattern of the nanowire region marked in **a**. **c**−**e** Transition from amorphous to crystalline diffraction patterns along the 20 nm Ni-BMG nanowires formed outside of the homogeneous flow region. **c** SAED over nanowire reveals the presence of crystalline phases. **d** HRTEM image of marked region in **c**. **e** FFT patterns from marked regions in **d** reveals a transition of the structure from amorphous to crystalline along the growth direction of the nanowire.

selected area electron diffraction (SAED; Fig. 3a) and the high-resolution TEM (HRTEM) image with fast Fourier transformation (FFT; Fig. 3b). On the other hand, crystallization sets in for the nanowire formed outside of the homogeneous flow region, which can be concluded from the distinguished spots in the SAED pattern (Fig. 3c and Supplementary Fig. 7). The FFT of high-resolution TEM carried out at various locations along the nanowire reveals crystallization (Fig. 3d, e). At the nanowire's root, which is close to the feedstock, the nanowire remains amorphous (Fig. 3e(1) and (2)). Upon further growth (increasing depth), the nanowire starts to crystallize (Fig. 3e(3)) and the crystallized volume fraction increases (Fig. 3e(4)). These changes in the atomic structure can be expected to result from the corresponding shifts in the chemical composition, and will be discussed in details later.

## Discussion

Our experimental findings provide evidence for a mechanism transition in the deformation behavior of amorphous metals, characterized by a deviation from the scaling behavior of homogeneous flow (Fig. 1). While enhanced deformation in amorphous metals can in theory be the result of various reasons, the observed scaling of $L$ vs. $d$, with an increasing deformation rate with decreasing length scale (Fig. 1d−f), suggests a mechanism based on surface process. This is due to the fact that as the surface-to-volume ratio increases with decreasing length scale, surface effects become pronounced and can eventually dominate the material properties. This principle underlies most of the size effects observed in nanomaterials. Fast atomic transport on the surface has been widely recognized to be the deformation mechanism in crystalline materials in a variety of scenarios, including grain boundary diffusion in high temperature creep and surface and interface diffusion in tension and compression of nanomaterials[24–26]. Fast surface transport was also reported in amorphous metals[27–29]. Therefore, we argue that when amorphous materials are deformed at very small length scale, as viscous flow is limited by non-sliding boundary condition and the surface ratio is

enlarged, transport along the surface/interface surpasses viscous flow as the dominant deformation mechanism.

Here, we consider two potential mechanisms controlling deformation of amorphous metals at the nanoscale: viscous flow and atomic diffusion (Fig. 4a). Theoretical models predict the two mechanisms' contrasting scaling behaviors between $L$ and $d$ (Supplementary Note 1 and 3). In the case of viscous flow (Eq. (1)), the length of the nanowires is proportional to the molding diameter ($L \propto d$; Fig. 4a and Supplementary Note 1). On the other hand, atomic diffusion typically utilizes the nanowire-mold interface as a diffusion path, where the interface diffusivity ($D_I$) can be several orders of magnitude higher than the bulk diffusivity[27,30]. This interface process causes $L$ to increase with decreasing $d$ (Fig. 4a) and takes the specific form (Supplementary Note 3):

$$L = \sqrt{\frac{8\delta D_I p \Omega t}{d k_B T}}, \qquad (2)$$

where $\delta$ is the thickness of the interface layer, $\Omega$ is the average atomic volume, and $k_B$ is the Boltzmann constant. Under fixed molding conditions, Eq. (2) gives $L \propto d^{-1/2}$ for diffusion, resulting in a qualitatively different scaling behavior than the scaling suggested by the viscous flow mechanism.

Based on this analysis, when amorphous metals are deformed at larger length scale, viscous flow dominates the deformation process and $L$ follows a linear scaling with $d$. Upon reduction of the sample size, as the flux based on homogeneous flow is limited by viscosity, fast atomic transport through interface diffusion takes over, and eventually becoming the dominant deformation mechanism. This results in a scaling of $L$ with $d^{-1/2}$ (Figs. 1 and 5). Our scaling experiments suggest such a competition and change in dominance between viscous flow and interface diffusion generally to be present in different amorphous metals (Fig. 1d−f) and under general molding conditions (Fig. 1 and Supplementary Fig. 2, also see in Fig. 5).

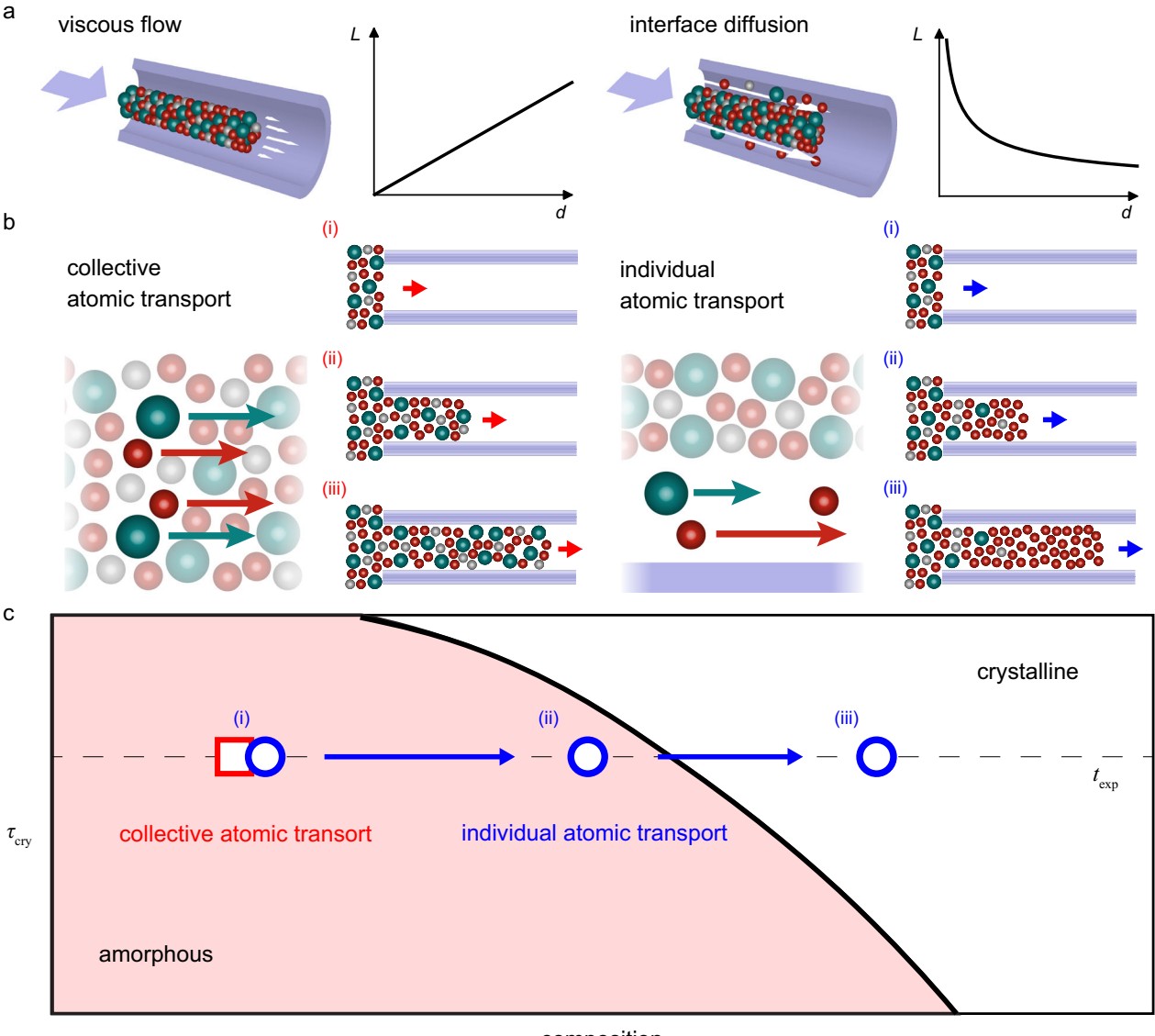

**Fig. 4 | Transition between collective and individual atomic transport. a** The two possible mechanisms for BMG deformation at the nano-scale, viscous flow, $L(d) \propto d$ (Eq. (1)) and interface diffusion, $L(d) \propto d^{-1/2}$ (Eq. (2)), where $L$ is the molding length and $d$ is the molding diameter, exhibit qualitatively and quantitatively different scaling of $L(d)$. **b** Within viscous flow, different elements have identical velocity of motion (depending on the location relative to the center of the nano-cavity), resulting in collective atomic transport with an unchanging and uniform composition along the nanowire. In contrast within the interface diffusion mechanism, elements move with different velocities, according to their different diffusivities in an individual atomic transport process. As more atoms of the faster diffuser reach the tip of the nanowire, the composition of the nanowire changes away from the nominal composition in the feedstock towards that of the faster diffuser. **c** For a highly optimized BMG composition like the here used $Ni_{45}Pd_{35}P_{16}B_4$, a change in composition generally shorten the time to reach crystallization ($\tau_{cry}$ ($T$, $c$)). Under conditions leading to collective atomic transport (red boxes, viscous flow), the BMG nanowire maintains its composition, hence leaving $\tau_{cry}$ unchanged. Therefore, if the processing time $t_{exp} < \tau_{cry}$, crystallization does not take place during deformation and an amorphous nanowire forms. Under conditions leading to deformation based on individual atomic transport, the composition changes which can lead to $\tau_{cry}$ ($c$) $< t_{exp}$, hence causing crystallization (blue circles, corresponding to (i) to (iii) in b).

Therefore, one plausible explanation for the experimental findings presented above is the transition in the deformation mechanism between viscous flow and interface diffusion. This transition is accompanied by a change in the chemical composition of the nanowires (Fig. 2). While atoms move collectively during viscous flow and their mobility is the same for all alloy constituents, during interface diffusion, atoms move individually according to their constituent-specific diffusivity. Consequently, this element-specific diffusion changes the chemical composition in the formed nanowires, whereas during viscous flow the composition remains unchanged (Fig. 4b). The observed change in the composition of formed Ni-BMG nanowires is in line with interface diffusion when considering the atomic sizes of Pd

(137 pm) and Ni (124 pm) which suggests Ni as the faster diffuser in the amorphous $Ni_{45}Pd_{35}P_{16}B_4$ (Fig. 4b).

The change in the nanowire composition formed through individual atomic transport can further affect the crystallization kinetics of the nanowire (Fig. 4c). The glass forming ability (GFA) of an alloy is generally a strong function of its composition[17,31–35]. For highly engineered BMG alloys, like the here considered $Ni_{45}Pd_{35}P_{16}B_4$, the GFA is optimized and can be expected to decrease when changing the composition away from the known BMG composition[33,34]. In other words, the time to reach crystallization ($\tau_{cry}$) is maximum at the known BMG composition and decreases with changing composition. If $\tau_{cry}$ falls below the experimental time scale ($t_{exp}$) due to changing composition,

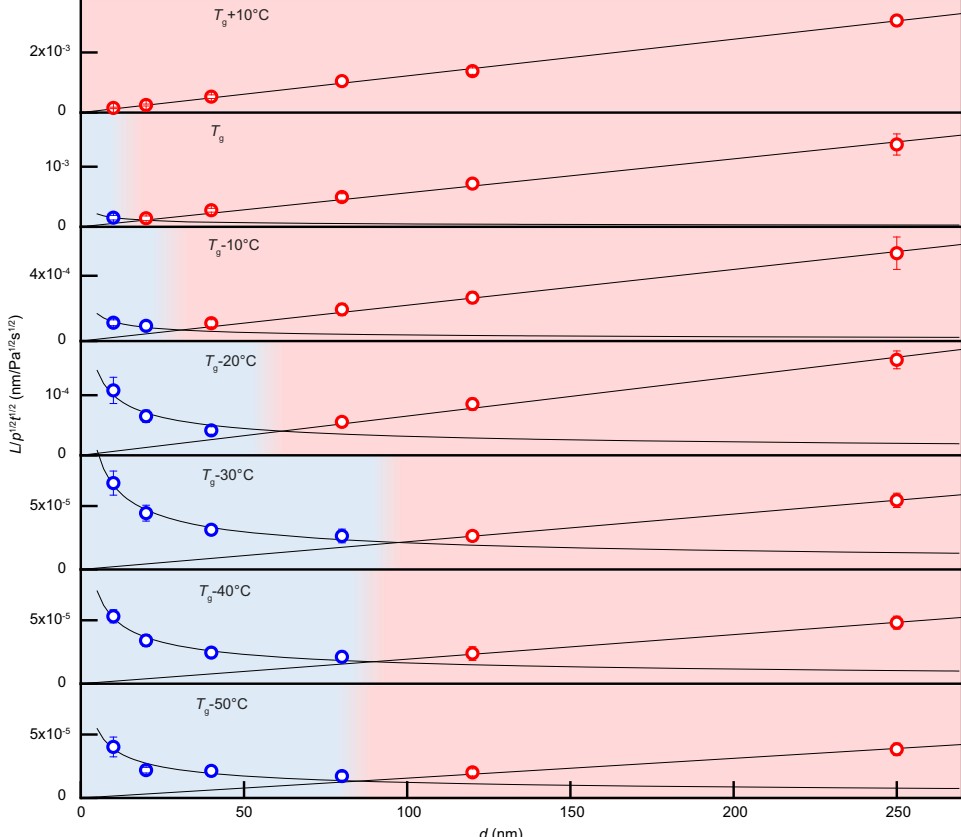

**Fig. 5 | Temperature dependency of the suggested deformation mechanism.**
The normalized length ($L/\sqrt{pt}$) vs. molding size ($d$) scaling suggests a transition in mechanism from collective atomic transport (viscous flow, red) to individual atomic transport (interface diffusion, blue) for temperatures from $T_g + 10\,°C$ to $T_g - 50\,°C$. The critical length scale of the transition, $d_c$, varies with temperature. For $T \geq T_g - 30\,°C$, $d_c$ increases with decreasing temperature, whereas for $T \leq T_g - 30\,°C$, $d_c$ decreases with decreasing temperature. All data points have error bars which are quantified by the standard deviation of more than 10 measurements. Source data are provided as a Source Data file.

crystallization occurs (Fig. 4c). During deformation based on individual atomic transport (Fig. 3c) the composition of the nanowires at the root is the same as the feedstock, hence exhibiting the largest $\tau_{cry}$, which exceeds $t_{exp}$ (blue (i) in Fig. 4c). With further growth of the nanowire, the composition changes and hence $\tau_{cry}$ decreases. For the structure to remain amorphous, some compositional changes can occur as long as $\tau_{cry} > t_{exp}$ (blue (ii) in Fig. 4c). However, upon further diffusion-controlled growth, the composition continues to change which can reach $\tau_{cry} < t_{exp}$, where the nanowire starts to crystallize (blue (iii) in Fig. 4c). Based on this mechanism, the length of the amorphous part in the nanowire is controlled by the differences among the diffusivities of the constituent elements. If the diffusivities vary largely among the constituents, the composition of the nanowire changes rapidly, which would lead to a short amorphous nanowire. On the other hand, if the diffusivities among the constituents are very similar, the composition of the nanowire changes very slowly and it can be expected that crystallization sets in at a longer nanowire length. The above suggested change in composition effect on crystallization is not present when nanowires are deformed by collective atomic transport through viscous flow. In this case, the forming nanowire maintains its composition of the highest glass forming ability (largest $\tau_{cry}$), ensuring $\tau_{cry} > t_{exp}$. As a result, the formed nanowire remains in its amorphous structure (Fig. 3a).

The above discussion of the composition and atomic structure further suggests the physical picture of a transformation of the two different deformation modes and their size dependence (Supplementary Note 4). Deformation is generally motivated by the release of internal energy induced by the applied external stress. Driven by this external stress, different deformation mechanisms compete, and the one with the fastest energy release rate (deformation rate) dominates[26]. This forms the fundamental principle for the transition in the deformation mechanism during nanomolding. While the viscous flow-based deformation rate increases with size (Eq. (1)), the interface diffusion-based deformation rate decreases (Eq. (2)). Their different size dependencies suggest a crossover of the dominant mechanism (Fig. 1d–f).

The deformation rates for both mechanisms are either controlled by viscosity or interface diffusivity, which are both functions of the molding temperature. Therefore, one should expect that the transition in the dominating mechanism should also be temperature dependent. To explore such possible temperature dependence of the change in mechanisms, we carried out nanomolding over a broad temperature range and determined the normalized molding length ($L/\sqrt{pt}$) as a function of $d$ (Fig. 5). We found that the transition in the deformation mechanism as a function of $d$ is also a function of temperature. At high temperature ($T_g + 10\,°C$), collective atomic transport (viscous flow) dominates the deformation at all length scales $\geq 10$ nm, while for $T \leq T_g$, to $T_g - 50\,°C$, a transition length scale ($d_c$) between collective (red) and individual atomic transport regions (blue) is suggested (Fig. 5). Considering the $T$ and $d$ dependence of $L$ reveals that $d_c$ is temperature dependent. At $T_g$, $d_c \sim 19$ nm and increases to ~31 nm at $T_g - 10\,°C$, ~62 nm at $T_g - 20\,°C$, and ~97 nm at $T_g - 30\,°C$. Surprisingly, the temperature-dependence of $d_c$ is not monotonic, as $d_c$ decreases with further decreasing temperature, to $d_c \sim 85$ nm at $T_g - 50\,°C$ (Fig. 5 and Supplementary Note 5).

To explain the non-monotonic temperature dependence of $d_c$, we calculate the viscosity and interface diffusivity with the data from

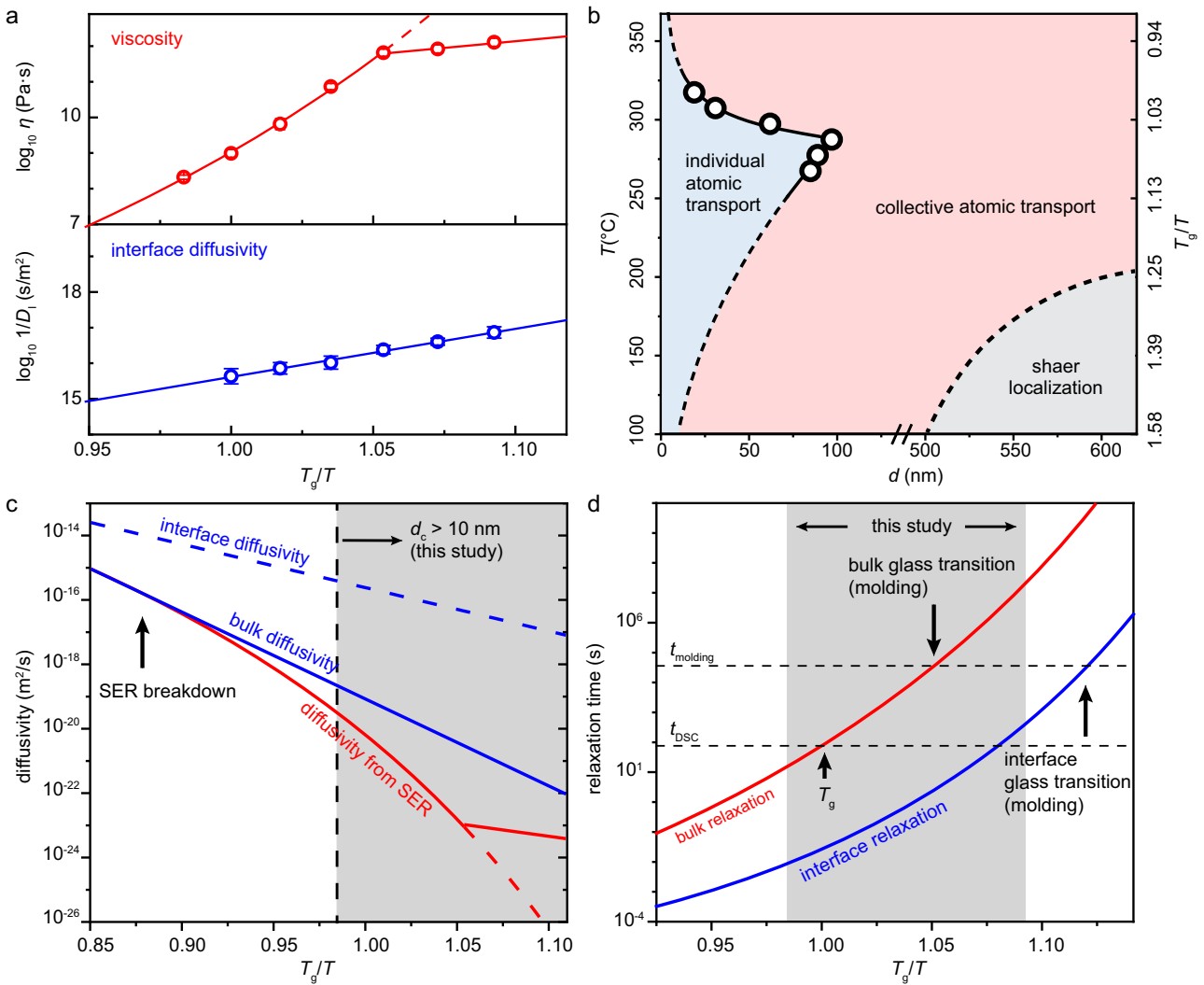

**Fig. 6 | Deformation mechanism map and material properties revealed by nanomolding. a** Viscosity (red) and interface diffusivity ($1/D_I$, blue) calculated according to $\eta = \frac{pt}{32}\left(\frac{d}{L}\right)^2$ and $D_I = \frac{L^2}{pt}\frac{dk_BT}{8\Omega\delta}$ for each temperature based on the experimental data in Fig. 5. A kink in the measured viscosity data at $T_g$ – 30 °C indicates the bulk glass transition under conditions used here for the nanomolding. The viscosity data of the metastable equilibrium state (supercooled liquid) is fitted with VFT equation, and for the glass state with an Arrhenius behavior. The interface diffusivity data are fitted with an Arrhenius behavior (solid blue line). For temperatures below $T_g$ – 30 °C ($\eta$ ~ $10^{12}$ Pa·s) the calculated viscosity no longer follows the VFT prediction, thereby inducing a non-monotonic behavior of the critical length scale, $d_c$. Source data are provided as a Source Data file. **b** Deformation mechanism map for nano-scale Ni-BMG based on the fitted viscosity and diffusivity

shown in **a**. The circles mark the $d_c$ measured from scaling experiments in Fig. 5. At even larger length scales than the here discussed collective to individual atomic transport transition, shear localization dominates the deformation at low temperatures when the experimental time scale exceeds the intrinsic relaxation time scale of the BMG. Source data are provided as a Source Data file. **c** Using the measured viscosity and interface diffusivity data, we calculated the bulk diffusivity of the Ni-BMG and revealed a breakdown of Stokes-Einstein relation (SER) at ~ 1.15 $T_g$. **d** Bulk and interface glass transition temperatures differ due to their different internal time scales[27, 30]. Considering the here used external time scales and using the experimentally determined bulk and interface relaxation times, the interface glass transition (with experimental time scale $t_{molding}$ = 36,000 s) occurs at ~ 254 °C and the bulk glass transition at ~ 288 °C.

nanomolding (Fig. 6a). Using the fitting result of interface diffusion and viscous flow segments, $\eta$ is obtained by $\eta = \frac{pt}{32}\left(\frac{d}{L}\right)^2$, and $D_I$ through $D_I = \frac{L^2}{pt}\frac{dk_BT}{8\Omega\delta}$ (Supplementary Note 6). Fig. 6a exhibits the so-calculated temperature-dependency of $\eta$ and $D_I$. A kink in the measured viscosity data at 287 °C ($T_g$ – 30 °C) indicates the glass transition present under the applied nanomolding conditions ($t_{exp}$ = 36,000 s)[36,37]. At temperatures above $T_g$ – 30 °C, $\eta$ increases rapidly following the Vogel-Fulcher-Tammann (VFT) equation of $\eta = \eta_0\exp\left(\frac{D^*_{VFT}T_0}{T-T_0}\right)$, where $D^*_{VFT}$ is a fragility parameter and $T_0$ is the temperature at which the relaxation time diverges. At $T \leq 287$ °C, the supercooled liquid freezes into a glass, evidenced by a change of fragility, $\frac{\partial \log(\eta)}{\partial T_g/T}$, which than follows Arrhenius behavior (Fig. 6a). In contrast, $D_I$ reveals a smooth function of temperature (Fig. 6a).

We argue that the non-monotonic behavior of $d_c$ originates from the different temperature behavior of $D_I$ and $\eta$. Assuming the fastest energy release rate is chosen, a transition in mechanism takes place where collective and individual atomic transport results in the same molding length. Under identical molding conditions ($p$, $t$, and $T$), Eqs. (1) and (2) give the molding length for collective atomic transport as $L_{col} = \sqrt{\frac{pt}{32\eta}}d$, and for individual atomic transport $L_{ind} = \sqrt{\frac{8p\Omega t}{k_BT} \cdot \delta D_I \frac{1}{d}}$. Equating $L_{col} = L_{ind}$, the corresponding critical length scale is (Supplementary Note 7)

$$d_c = \left(\frac{256\Omega}{k_BT}\delta\eta/(1/D_I)\right)^{1/3} \tag{3}$$

Equation (3) reveals that $d_c$ is proportional to $(\eta/(1/D_I))^{1/3}$, showing a similar non-monotonic behavior at 287 °C.

We use Eq. (3) to predict $d_c$ over a wide temperature range and summarize the results in a deformation mechanism map (Fig. 6b). Within the entire temperature range, individual atomic transport dominates the deformation at small scales, whereas collective atomic transport takes over at large scales. The range of diffusion-dominated deformation reaches a maximum, up to ~100 nm, at 287 °C. Since this temperature for maximum $d_c$ originates from the kink in the ratio between $\eta$ and $1/D_I$ due to the bulk glass transition, we can also expect this maximum temperature to shift with different experimental time scales ($t_{exp}$). At low temperatures ≤100 °C, the map suggests that individual atomic transport through interface diffusion can still dominate deformations at scales below 10 nm[28,29].

Deformation describes the transformation of a body from a reference configuration to the current configuration in response to an external stress. In this study, metallic glasses are plastically deformed into nanomolds by an applied compression pressure. The nano-features form from an originally flat surface. Plastic deformation dominated by atomic diffusion has been recognized in crystalline materials for high temperatures (e.g., Coble creep) or ultra-small sample sizes[24,25]. Nevertheless, such diffusion dominated deformation has not been observed in metallic glasses where homogenous deformation is envisioned as viscous flow. We argue that the suggested transition of deformation mechanism from collective (viscous flow) to individual atomic transport (diffusion) is due to i) the increasing effectiveness of interface diffusion when the surface ratio is increased by reducing sample size, and ii) the decreasing effectiveness of viscous flow when decreasing size due to boundary conditions which require higher and higher shear rates. The later factor is more pronounced in the here presented compression scenario. For the tension case, other boundary conditions such as the surface tension working against the sample elongation will replace the here present non-sliding boundary condition in the compression case. As a result, a similar mechanism transition will be present in tension, however at a different critical dimension than in the compression case.

The here uncovered transition from collective to individual transport also reveals the breakdown of the Stokes-Einstein relation (SER)[38,39]. At high temperatures in the liquid state, viscosity and bulk diffusivity ($D_B$) are quantitively correlated through the Stokes-Einstein relation, $D_B \eta = \frac{k_B T}{c \pi R}$, where $R$ is the effective radius of the diffusing particle and c is a constant. This relation decouples when the temperature approaches $T_g$, and the change of $\eta$ and $D_B$ with temperature are independent. A deviation between the temperature dependent curves of $D_B$ and the diffusivity from SER ($D_{SER} = \frac{k_B T}{c \pi R} \frac{1}{\eta}$, as a function of viscosity) indicates the breakdown temperature of SER (Fig. 6c). Therefore, this breakdown temperature ($T_{SER}$) can be revealed by measuring the temperature dependent viscosity and bulk diffusivity. To extract the bulk diffusivity from nanomolding, we consider the random first-order transition theory and set the activation energy for bulk diffusion $Q_B = 2Q_I = 3.16$ eV[30,40]. Here, the activation energy for the interface diffusion ($Q_I$) of the Ni-BMG is measured through nano-molding (Fig. 6a and Supplementary Note 6). The Arrhenius plot of $D_B$ follows the curve of $D_{SER}$ at $T_{SER} = 1.15 T_g$ ~ 408 °C (Fig. 6c, and details in Supplementary Note 8). This breakdown temperature of the Stokes-Einstein relation is similar to direct measurement with radiotracer methods from similar glass forming systems[38]. It should be mentioned that the above assumption of $Q_B = 2Q_I$ is a rough estimation. The predicted absolute value of temperatures with SER breakdown and some other materials properties may shift with the actual value of $Q_B$ (Supplementary Note 9).

Under collective transport, the temperature dependent viscosity exhibits a kink at 287 °C, which we consider the bulk glass transition in nanomolding[36,37]. However, a similar kink does not exist for interface diffusivity measured through individual atomic transport (Fig. 6a). This

indicates a different glass transition temperature at the interface than in the bulk[27]. Glass transition occurs at the temperature where the relaxation time scale ($\tau$) equals the experimental time scale ($t_{exp}$), $\tau = t_{exp}$. In amorphous metals, higher atomic mobility on the surface/interface results in shorter relaxation time than in bulk and, consequently, a lower glass transition temperature[27,30,41–43]. Using the relaxation time determined by differential scanning calorimetry at $T_g$ of $\tau_g = 76$ s, we achieved a VFT fitting of the bulk relaxation time ($\tau_B$) by $\tau_B \propto \eta$ (Fig. 6c, d and Supplementary Note 9), and interface relaxation time ($\tau_I$) by $\tau_I = \tau_B \frac{D_I}{D_B}$ (Fig. 6d and Supplementary Note 9). For the here used conditions for nanomolding of $t_{exp} = 36000$ s, these relaxation times result in a glass transition at the interface at 254 °C and in the bulk at 288 °C. The latter is very similar to the kink in the temperature dependent viscosity at 287 °C. The interface glass transition temperature is ~30 °C lower than that of the bulk, comparable to values determined through electron correlation microscopy and simulation results of other metallic glass systems[27,44].

Beyond its scientific relevance, the suggested change in deformation mechanism with decreasing size towards individual atomic transport may also suggest novel synthesis methods for metallic glasses. The individually diffusing atoms diffuse down the pressure gradient until they reach the tip of the nanowire[45–47]. Here, they are templated on the amorphous structure and assume a position in an epitaxial-manner. This mechanism results in packing motifs of forming nanowire only occurring on the 2D surface, different from the typical volume process where 3D packing motifs dominate (such as widely used melt-quenching methods)[48]. Such different packing may result in different glass structures. As discussed above (Fig. 3), the different diffusivities of the alloy constituents cause the composition of the forming nanowire to change. Such changing composition generally reduce glass forming ability and the growth of the amorphous nanowire is terminated when the compositional changes induce crystallization (Fig. 3c–e). As the difference between the diffusivities of the constituents determines how rapidly composition change, it is reasonable to assume that with BMG alloys made from constituents with similar diffusivities, long amorphous nanowires can be epitaxially grown.

In conclusion, we have uncovered a size-dependent transition in the deformation behavior in nanosized amorphous metals using nanomolding at elevated temperatures. The deformation follows the widely accepted homogeneous flow at large scale, but deviates from such scaling for sample sizes below a critical length scale. This transition has been substantiated through scaling experiments and composition characterization which have revealed that deformation in amorphous metals is size- and temperature-dependent, which is summarized in a nano-scale deformation mechanism map. The experimental findings and theoretical framework established in this work suggest that, at all temperatures, collective atomic transport through viscous flow dominates at large sizes, while individual atomic transport through interface diffusion takes over at small scales. The critical length scale of the transition between individual and collective atomic transport is a non-monotonic function of temperature, exhibiting a maximum at the glass transition. Nanowire formed through individual atomic transport exhibit a gradient in chemistry, which can also lead to crystallization. If crystallization can be avoided, forming nanowires through individual atomic transport may suggest novel synthesis methods for amorphous metals. Further, the ability to realize conditions where these two transport mechanisms are comparable also suggests insights into nano-mechanics and the physics of amorphous metals, beyond the here revealed breakdown of the Stokes-Einstein relation and the difference in bulk and surface glass transition temperatures.

## Methods

### Preparation of BMGs

The $Ni_{45}Pd_{35}P_{16}B_4$ BMG used in this study is a composition that we developed which is inspired by previous researches[49]. We prepared the

$Ni_{45}Pd_{35}P_{16}B_4$ and $Pd_{43}Ni_{10}Cu_{27}P_{20}$ BMG through melt-quenching. A master alloy (~10 g) was prepared by induction melting of high purity raw materials of Ni, Pd, $Ni_{50}P_{50}$ alloy, Cu, and B (>99.95 %) in an evacuated quartz tube (with vacuum level <10 mTorr). We then fluxed the alloy with dehydrated boron trioxide ($B_2O_3$) at 1150 °C for 30 min. Melt-quenching was then applied by re-melting the fluxed alloy at 1100 °C for 2 min in vacuum and subsequent 1 min in Argon atmosphere (positive pressure of ~$5 \times 10^5$ mTorr) before vitrified into a 3 mm rod by water quenching. $Zr_{46}Cu_{46}Al_8$ BMG was fabricated through suction casting. A master alloy (~5 g) was mixed by arc melting from high purity constituent (99.99+ %, from Alfa Aesar) and in positive Argon pressure. The alloy was melted for at least six times and turned over between melting to ensure fully mixing. This master alloy was then remelted in positive Argon pressure and suction casted into a water-cooled cylinder copper mold with 2 mm as the inner diameter. The fabricated BMG rods were sliced into discs of thickness ~ 0.8 mm with surfaces polished down to ~ 1 μm for nanomolding. The amorphous nature of the prepared BMG feedstocks was confirmed by X-ray diffraction (XRD) and differential scanning calorimetry (DSC) (Supplementary Figs. 1 and 8).

### Nanomolding

Ni-based, Zr-based, and Pd-based BMG feedstocks were deformed by nanomolding with various molding conditions. Finely polished feedstock discs were first prepressed against the mold (porous anodic aluminum oxide, AAO) of desired diameter ($d$) at 337 °C for Ni-BMG (20 °C above calorimetric $T_g$) with a linearly increasing load from 0 to 350 MPa in 30 s and hold for 3 s before releasing, 470 °C for Zr-BMG (30 °C above calorimetric $T_g$) with a linearly increasing load from 0 to 280 MPa in 30 s and hold for 30 s before releasing, and 330 °C for Pd-BMG (20 °C above calorimetric $T_g$) with a linearly increasing load from 0 to 560 MPa in 30 s and hold for 3 s before releasing. These conditions were carefully calibrated to sufficiently remove the surface roughness of the feedstock and result in only small prints from the mold with aspect ratio smaller than 0.5 (Supplementary Fig. 8). The BMG-mold combination was then tuned to the molding temperature ($T$) and molded with controlled pressure ($p$) and time ($t$) without demolding. We adjusted the molding conditions of $p$ and $t$ for different molding temperatures to get practical length of nanowires for length measurement. These conditions are: for Ni-BMG $p = 1$ GPa, $t = 300$ s for 327 °C and 317 °C; $p = 2$ GPa, $t = 900$ s for 307 °C; $p = 1.25$ GPa, $t = 36000$ s for 297 °C; and $p = 2$ GPa, $t = 36,000$ s for 287 °C, 277 °C, and 267 °C, for Zr-BMG $p = 1.25$ GPa, $t = 900$ s for 420 °C, and for Pd-BMG $p = 2$ GPa, $t = 3600$ s for 290 °C. Upon nanomolding, the AAO mold was removed by wet etching in 3 mol/L potassium hydroxide (KOH) solution at room temperature for 10 hours followed by rinsing in distilled water for 10 hours. We then characterized the length of molded nanowires by scanning electron microscopy (SEM).

### Composition and atomic structure characterizations

The nanostructure of the molded rods was characterized using transmission electron microscopy (TEM, Thermo Fisher Scientific, Talos F200X and Tecnai F20 G2) equipped with a Super-X energy-dispersive X-ray spectrometer (EDS). It is essential to prepare the TEM samples including the BMG feedstock and the deformed nanorods altogether to see the change of chemical composition and crystal structure. Thus, the thin lamellas for TEM were carefully prepared in focused ion beam (FIB, Hitachi, NX5000) keeping the original structure. Lamellas were taken from inside of the mold combinations after mechanical polishing the cross-section edge of the combinations followed by an additional 100 μm milling by Ga+ ion in FIB (Supplementary Fig. 9(i) and (ii)). For a uniform fine milling, the lifted lamella was rotated 90 degrees so that the BMG feedstock was top layer of the sample along ion milling direction while the porous AAO template went down, and then carbon was deposited as protection layer on the BMG (Supplementary Fig. 9(iii) and (iv)). These lamellas were carefully thinned down to a thickness similar to the corresponding nanowire diameters ($d = 20$ nm and $d = 120$ nm). To prevent the lamella bending which results in decoupling of the BMG feedstock and the AAO template, only the upper part of the sample was further milled leaving the lower part to support the lamella rigidly (Supplementary Fig. 9(v)).

## Data availability

All data needed to evaluate the conclusion in this paper are present in the paper and/or the Supplementary Information. Source data are provided with this paper.

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

## Acknowledgements

This work was supported by the National Science Foundation through the Advanced Manufacturing Program (CMMI1901613). H. J. Chang and M. Y. Na were supported by the Ministry of Trade, Industry, and Energy (MOTIE) of Korea through the project No. P0022331 supervised by the Korea Institute for Advancement of Technology (KIAT).

## Author contributions

N.L., S.S., and J.S. conceived the idea and designed the experiments. N.L. conducted the nanomolding and the data processing. M.Y.N., G.H.P., and H.J.C. conducted the FIB-TEM characterization of composition and atomic structures. N.L., A.R., G.L., and S.A.K. conducted the length characterization of nanowires. S.S., F.Y., and Y.L. prepared the MG samples. N.L., S.S., and J.S. analyzed the data and wrote the manuscript. All authors discussed the manuscript. J.S. supervised the project.

## Competing interests

The authors declare no competing interests.
