## [Peer Review File · Nature Communications]

Size-dependent Deformation Behavior in Nanosized Amorphous Metals Suggesting Transition from Collective to Individual Atomic TransportEditorial Note: This manuscript has been previously reviewed at another journal that is not operating a transparent peer review scheme. This document only contains reviewer comments and rebuttal letters for versions considered at *Nature Communications*.

REVIEWER COMMENTS

Reviewer #1 (Remarks to the Author):

This is the third round of review. We have not made much progress with the debate in question, as I still remain unpersuaded with any of the arguments presented by the authors, specifically with the mechanism governing the reported observation. After two review rounds, all my concerns still hold despite the authors' extensive responses. I can go over their response and refute all arguments, one by one, as I did in my last review. But I don't believe this is any longer constructive and will not serve science.

The authors correctly point out that our disagreement is not with the experimental finding but the scientific interpretation of the finding. I do find the experimental finding new, interesting, and worthy of publication in an advanced journal. I also value the experimental work performed by the authors to be of very high quality, and I recognize the tremendous effort that went in to conduct such work. But the authors are not just reporting the experimental finding, which is not in dispute and could very well serve as the principal standalone finding. They are principally claiming discovery of a novel thermodynamic mechanism. But in reality, such mechanism is merely a result of the authors' own interpretation of their data rather than an experimentally observed mechanism. In my opinion, this interpretation is deeply controversial and questionable (if not fundamentally flawed). At best, the claimed mechanism is non-conclusive.

The authors observed enhanced filling accompanied by a chemical/compositional heterogeneity when nanomolding metallic glasses. In and of itself this observation is scientifically interesting. This is an experimental fact. The authors interpret data from this finding through a novel thermodynamic concept that assumes diffusion in the absence of flow. This is not an experimental fact. The authors offer no experimental evidence that directly corroborates this concept. Such evidence would be direct experimental observation/visualization supporting filling of the channel by atomic diffusion from the channel interface in the absence of flow. Rather, the authors claim this mechanism as a scientific fact merely because it is theoretically consistent with their data fitting. Unless direct mechanistic evidence is presented to experimentally validate this mechanism, this mechanism remains a mere hypothesis. In my personal view (and I expect other members of our community will share this view), this hypothesis is unfounded despite appearing to conform with the authors analysis. I brought up a different mechanism that also explains the data, strain-rate induced de-mixing, as an alternative and more plausible hypothesis. The authors dismissed this mechanism not by presenting experimental data corroborating their claimed mechanisms but by a process of elimination involving a series of hypothetical arguments

and crude fitting approximations. This is not exactly scientific.

Nonetheless, as I mentioned above, I believe it would be counterproductive to continue debating which hypothesis is most likely to be at play in the absence of direct experimental mechanistic evidence. I also believe the authors have earned the right to publish their experimental finding. They may also present an interpretation that in their opinion is the prevailing hypothesis explaining their experimental finding, so long as this interpretation is not presented as an experimental discovery. The community will ultimately be the fair judge of this hypothesis. Currently, the manuscript presents the authors' interpretation of the data as an experimental fact. Thus, it is unacceptable in its current version. If the authors revise and reorganize the manuscript content to clearly separate the actual experimental facts from the theoretical interpretations, I would be willing to recommend the manuscript for publication in Nature Communications.

Specifically, in my view the title, abstract, and conclusions, but also the body of the manuscript should all be revised to reflect the enhanced filling and chemical heterogeneity observed at nanosized metallic glasses as the core finding, rather than presenting the "collective to individual atomic transport" mechanism as a key discovery. This proposed mechanism should instead be introduced as a prevailing hypothesis found by the authors to be consistent with the data, after also having considered other mechanisms (e.g. strain-rate induced de-mixing or other). For example, the title "Transition from Collective to Individual Atomic Transport in Nanosized Amorphous Metals" is misleading. So is the sentence in the abstract: "Here we reveal that the underlying deformation mechanism of amorphous metals changes at small scales, typically below 100 nm, from collective atomic transport through viscous flow to an individual atomic transport based on atomic diffusion", or in the conclusions: "In conclusion, we have uncovered a deformation mechanism based on interface diffusion in amorphous metals using nanomolding." These are just three examples with which I believe the authors mislead the reader by offering their interpretation as an experimental fact. Many more such claims appear throughout the manuscript and need to be reversed. This is the only way I can see the manuscript conveying the interesting experimental finding without conflating the theoretical interpretation as an experimentally corroborated discovery.

Lastly, I'd like to say a final word on scaling. I strongly disagree with the authors that scaling does not require non-dimensionality. Scaling absolutely requires non-dimensionality, as scaling is the purpose and outcome of dimensional analysis. The authors should not base their perception on one-line snippets from Google but rather study the concepts of dimensional analysis, self-similarity, and scaling from fundamental textbooks. One reference that comes to mind is chapter 3.7 from textbook "Transport Phenomena" by Bird, Stewart, and Lightfoot (2nd Ed., 2002, p. 97). Other chapters like this can be found in various textbooks on transport phenomena or critical phenomena. I hope the authors reconsider their theoretical construction and revise it to be consistent with the accepted fundamentals of dimensional analysis.

Response to the Review Comments

Reviewer #1 (Remarks to the Author):

This is the third round of review. We have not made much progress with the debate in question, as I still remain unpersuaded with any of the arguments presented by the authors, specifically with the mechanism governing the reported observation. After two review rounds, all my concerns still hold despite the authors' extensive responses. I can go over their response and refute all arguments, one by one, as I did in my last review. But I don't believe this is any longer constructive and will not serve science.

The authors correctly point out that our disagreement is not with the experimental finding but the scientific interpretation of the finding. I do find the experimental finding new, interesting, and worthy of publication in an advanced journal. I also value the experimental work performed by the authors to be of very high quality, and I recognize the tremendous effort that went in to conduct such work. But the authors are not just reporting the experimental finding, which is not in dispute and could very well serve as the principal standalone finding. They are principally claiming discovery of a novel thermodynamic mechanism. But in reality, such mechanism is merely a result of the authors' own interpretation of their data rather than an experimentally observed mechanism. In my opinion, this interpretation is deeply controversial and questionable (if not fundamentally flawed). At best, the claimed mechanism is non-conclusive.

The authors observed enhanced filling accompanied by a chemical/compositional heterogeneity when nanomolding metallic glasses. In and of itself this observation is scientifically interesting. This is an experimental fact. The authors interpret data from this finding through a novel thermodynamic concept that assumes diffusion in the absence of flow. This is not an experimental fact. The authors offer no experimental evidence that directly corroborates this concept. Such evidence would be direct experimental observation/visualization supporting filling of the channel by atomic diffusion from the channel interface in the absence of flow. Rather, the authors claim this mechanism as a scientific fact merely because it is theoretically consistent with their data fitting. Unless direct mechanistic evidence is presented to experimentally validate this mechanism, this mechanism remains a mere hypothesis. In my personal view (and I expect other members of our community will share this view), this hypothesis is unfounded despite appearing to conform with the authors analysis. I brought up a different mechanism that also explains the data, strain-rate induced de-mixing, as an alternative and more plausible hypothesis. The authors dismissed this mechanism not by presenting experimental data corroborating their claimed mechanisms but by a process of elimination involving a series of hypothetical arguments and crude fitting approximations. This is not exactly scientific.

Nonetheless, as I mentioned above, I believe it would be counterproductive to continue debating which hypothesis is most likely to be at play in the absence of direct experimental mechanistic evidence. I also believe the authors have earned the right to publish their experimental finding. They may also present an interpretation that in their opinion is the prevailing hypothesis explaining their experimental finding, so long as this interpretation

is not presented as an experimental discovery. The community will ultimately be the fair judge of this hypothesis. Currently, the manuscript presents the authors' interpretation of the data as an experimental fact. Thus, it is unacceptable in its current version. If the authors revise and reorganize the manuscript content to clearly separate the actual experimental facts from the theoretical interpretations, I would be willing to recommend the manuscript for publication in Nature Communications.

Specifically, in my view the title, abstract, and conclusions, but also the body of the manuscript should all be revised to reflect the enhanced filling and chemical heterogeneity observed at nanosized metallic glasses as the core finding, rather than presenting the "collective to individual atomic transport" mechanism as a key discovery. This proposed mechanism should instead be introduced as a prevailing hypothesis found by the authors to be consistent with the data, after also having considered other mechanisms (e.g. strain-rate induced de-mixing or other). For example, the title "Transition from Collective to Individual Atomic Transport in Nanosized Amorphous Metals" is misleading. So is the sentence in the abstract: "Here we reveal that the underlying deformation mechanism of amorphous metals changes at small scales, typically below 100 nm, from collective atomic transport through viscous flow to an individual atomic transport based on atomic diffusion", or in the conclusions: "In conclusion, we have uncovered a deformation mechanism based on interface diffusion in amorphous metals using nanomolding." These are just three examples with which I believe the authors mislead the reader by offering their interpretation as an experimental fact. Many more such claims appear throughout the manuscript and need to be reversed. This is the only way I can see the manuscript conveying the interesting experimental finding without conflating the theoretical interpretation as an experimentally corroborated discovery.

Lastly, I'd like to say a final word on scaling. I strongly disagree with the authors that scaling does not require non-dimensionality. Scaling absolutely requires non-dimensionality, as scaling is the purpose and outcome of dimensional analysis. The authors should not base their perception on one-line snippets from Google but rather study the concepts of dimensional analysis, self-similarity, and scaling from fundamental textbooks. One reference that comes to mind is chapter 3.7 from textbook "Transport Phenomena" by Bird, Stewart, and Lightfoot (2nd Ed., 2002, p. 97). Other chapters like this can be found in various textbooks on transport phenomena or critical phenomena. I hope the authors reconsider their theoretical construction and revise it to be consistent with the accepted fundamentals of dimensional analysis.

Re: We acknowledge the reviewer's concern for scientific rigor and the importance of basing conclusions on experiments and logic. We share this perspective and have carefully examined the logic behind our conclusions.

We had several meetings and went through and evaluated the logic of the conclusion that among today's understanding other possibilities can be ruled out as they contradict (at least partially) the experiments. Yes, there is a possibility that future concepts may reveal flaws or even an alternative interpretation of the experimental finding. Based on this very careful carried out scientific study, which has been meticulously designed in its evolution by comparing against existing theories and interpretations the only explanation that does not contradict some experimental evidence is the transition from collective to individual transport. As we at length discussed in our last response letter, strain rate induced change

in composition and subsequent crystallization is inconsistent with our experimental findings. Throughout our research discussions, we have not identified any other concept that can adequately describe our results. Therefore, we respectfully disagree with the notion of alternative interpretations of our data. However, in the interest of progressing and scientific discourse, we are willing to adopt a very cautious approach as suggested by the reviewer.

As such, we have made significant revisions to the manuscript to avoid presenting our conclusions as absolute facts, but rather as possibilities. We believe that this aligns with the reviewer's intent.

Specifically, we have followed the reviewer's recommendation to separate the experimental results from our interpretation of the transition from collective to individual deformation as a function of sample size. We have to admit that some of the changes have been painful (as we clearly see the correlation). We have applied a cautious and logical approach to ensure the integrity of our findings and interpretations.

Particularly, the following aspects were revised:

We have made a thorough and clear separation between the theoretical framework from the experimental results section. All the analysis related to the interface diffusion mechanism and the transition between collective and individual atomic transport have been moved to the discussion section, after presenting all of the experimental facts. According to the separation in the text, we have also removed the previous Fig. 1c, Fig. 2a, and Fig. 3a, which discussed the collective and individual atomic transport from Fig. 1, 2, and 3, and combined them into a new Fig. 4 at the beginning of the discussion section. The separated text of the experimental facts and the theoretical interpretation have been further revised to ensure a logical flow in the manuscript.

We clearly stated that the collective to individual atomic transport is “one plausible explanation for the experimental findings” (in the discussion section), and also toned down through out the manuscript by indicating the experimental facts “suggest” or “can be explained by” the proposed transition in atomic transport as one possibility.

We revised the title of the manuscript according to the reviewer's suggestion to emphasize the experimental facts and decouple them from the interpretation. The revised title is:

“Size-dependent Deformation Behavior in Nanosized Amorphous Metals Suggesting Transition from Collective to Individual Atomic Transport”

We revised the abstract to clearly identify the experimental findings, such as the size-dependent transition in deformation behavior and the composition shift, and the interpretation, which focuses on the transition in atomic transport, separately. The revised abstract is attached as below:

“The underlying atomistic mechanism of deformation is a central problem in mechanics and materials science. Whereas deformation of crystalline metals is fundamentally understood, the understanding of deformation of amorphous metals lacks behind, particularly identifying the involved temporal and spatial scales. Here, we reveal that at small scales the size-dependent deformation behavior of amorphous metals significantly deviates from homogeneous flow, exhibiting increasing deformation rate with reducing size and gradually shifted

composition. This transition suggests the deformation mechanism changes from collective atomic transport by viscous flow to individual atomic transport through interface diffusion. The critical length scale of the transition is temperature dependent, exhibiting a maximum at the glass transition. While viscous flow does not discriminate among alloy constituents, diffusion does and the constituent element with higher diffusivity deforms faster. Our findings yield new insights into nano-mechanics and glass physics, and may suggest alternative processing methods to “epitaxially” grow metallic glasses.”

We revised the introduction and conclusion in a similar way as above by clearly distinguishing the experimental facts from the interpretation. Specifically, the following sentences are added/modified:

(Introduction) “... Our research reveals a notable size-dependent transition in the deformation behavior of nanosized amorphous metals under compression, occurring at a length scale typically around or below 100 nm. This critical length scale is dependent on both the temperature of deformation and the specific material under study. For sizes larger than the critical length scale, deformation follows the expected description of viscous flow and increasing size resulting in faster deformation rate. However, for sizes below the critical length scale, a substantial deviation from the homogeneous viscous flow model is observed, exhibiting an increasing deformation rate with reducing size, along with a gradual shifting in the composition of the deformed material. This different scaling behavior of deformation with respect to size indicates a transition in the underlying deformation mechanism, which can be quantitatively explained by the competition and change in dominance between collective atomic transport through homogeneous viscous flow and individual atomic transport via interface atomic diffusion. The experimental findings presented in this work, along with the theoretical framework established, provide novel insights into nano-mechanics and physics of amorphous materials, ...”

(Conclusion) “In conclusion, we have uncovered a size-dependent transition in the deformation behavior in nanosized amorphous metals using nanomolding at elevated temperatures. The deformation follows the widely accepted homogeneous flow at large scale, but deviates from such scaling for sample sizes below a critical length scale. This transition has been substantiated through scaling experiments and composition characterization which has revealed that deformation in amorphous metals is size- and temperature- dependent, which is summarized in a nano-scale deformation mechanism map. The experimental findings and theoretical framework established in this work, suggest that for all temperatures, collective atomic transport through viscous flow dominates at large sizes, while individual atomic transport through interface diffusion takes over at small scales....”

Furthermore, other details that have been revised alongside the major changes above are highlighted in the revised text.

As to the scaling law presented in this study, we realized that there might have been some unexpected confusions in the terminology within the previous rounds of review and

response. The reviewer was citing (“Transport Phenomena” by Bird, Stewart, and Lightfoot) and referring to “scaling”, which is widely used in mechanics (especially fluid mechanics) and engineering, meaning to approximate the properties of a full-scale system of interest from a (usually) smaller scale model system where experimental analysis is designed and acquired. One example for this “scaling” is wind tunnel experiments where aerodynamical data is achieved and “scaled-up” to full scale airplanes. “Scaling” requires geometric similarity and dynamic similarity, which, as the reviewer mentioned, are from dimensional analysis and based on dimensionless numbers. Different to this definition above, in the current study and our previous responses we are referring to “scaling law” that is also a widely used concept in literatures. “Scaling law” or “scaling behavior” describes the way one physical value changes with another (when all other affecting parameters are fixed). One example is the gravity follows a power scaling law of the distance, e.g., $F \propto r^{-2}$. Scaling laws study the absolute value of physical quantities, which can be both dimensional and dimensionless numbers. And the scaling behavior revealed in the current study belongs to the dimensional case. The definition of “scaling law” that we cited in our last response is from the subject definition of Nature Portfolio, and we think it should be suitable to use in this particular situation. However, we feel sorry for this confusion in terminology and think it might help by modifying the L/\sqrt{pt} axis in Fig. 1 that is not a commonly used way to show scaling behavior of molding length. Therefore, we have changed the y-axis from L/\sqrt{pt} to L in Fig. 1d, e, and f in this revised manuscript.

With the revisions above, the manuscript now clearly separates the actual experimental results from the theoretical interpretation. We believe that this revised version efficiently addresses the reviewer’s comments.